# MicroRNA-21 Protects Hypoxic-Induced Cardiomyocytes Injury by Targeting Smad-7

**DOI:** 10.3390/cells14191483

**Published:** 2025-09-23

**Authors:** Md Sayed Ali Sheikh, A. Alduraywish, Basil Mohammed Alomair, Muhannad Almubarak, Umme Salma

**Affiliations:** 1Internal Medicine Department, College of Medicine, Jouf University, Sakaka 72345, Saudi Arabia; dr-aaad@ju.edu.sa (A.A.); bmalomair@ju.edu.sa (B.M.A.); maalmubarak@ju.edu.sa (M.A.); 2Gynecology and Obstetrics Department, College of Medicine, Jouf University, Sakaka 72345, Saudi Arabia; dusalma@ju.edu.sa

**Keywords:** microRNAs, acute myocardial infarction, biomarkers, H9c2 cells, Smad-7, aging

## Abstract

Globally acute myocardial infarction is the leading independent cause of unexpected death. This study aimed to explore the diagnostic and molecular impact of miR-21, miR-488, and miR-126 in acute myocardial infarction patients (AMI). We enrolled 95 non-ST-elevation myocardial infarction (NSTEMI) patients, 152 ST-elevation myocardial infarction (STEMI) patients, and 95 healthy individuals, additionally using three-month-old mice and their ventricular-derived H9c2 cells. The circulatory plasma miR-21 and miR-488 levels were significantly upregulated, while plasma miR-126 levels were remarkably downregulated in NSTEMI and STEMI subjects. The receiver operating characteristic curve showed that plasma miR-21, miR-488, and miR-126 were able to clearly differentiate NSTEMI and STEMI from healthy subjects. Moreover, H9c2 hypoxic cells treated with inhibitor miR-21 markedly reduced intracellular ROS levels, capase-3 activities levels, and cellular apoptosis rates and significantly enhanced cellular viability through up regulation of Smad-7 mRNA and protein expressions. In geriatric STEMI and NSTEMI subjects, plasma miR-21 levels were evidently higher than in comparatively younger subjects. Upregulated plasma miR-21 and miR-488 levels and downregulated miR-126 levels might be considered potential clinical biomarkers for myocardial infarction patients, while inhibition of miR-21, which significantly reduced hypoxia-exposed H9c2 cellular injury via targeting Smad-7, could be a new therapeutic target for AMI patients. Low levels plasma miR-21 may have a significant impact on delaying the aging process.

## 1. Introduction

Acute myocardial infarction (AMI) is the number one cause of death and disability in both developed and developing countries of all races [1]. Early and correct diagnosis of AMI has a significant impact on the treatment and prognosis of the patient. The diagnosis of AMI is mainly based on clinical presentation, ECG changes, and biochemical markers. However, clinical presentation and ECG changes in elderly and diabetic patients are often nonspecific. Currently, cardiac troponins (cTnT or cTnI) are widely used biomarkers for the early evaluation of AMI in clinical practice. Moreover, several recent studies have found that cardiac troponins are not only elevated in AMI but also elevated in end-stage renal disease, severe heart failure, atrial fibrillation, septic shock, and chronic stable coronary artery disease, even in apparently healthy subjects. Therefore, a novel biomarker for early and reliable diagnosis of AMI is still lacking [1,2]. MicroRNAs (miRNAs) are small, evolutionarily conserved, non-coding RNA molecules that play an essential role in post-transcriptional gene expression through translational repression or mRNA destabilization. Moreover, miRNAs have been directly and indirectly associated with the pathogenesis of numerous cardiovascular diseases [3]. Various molecular mechanisms of AMI, such as platelet activation and aggregation in the arterial wall, coronary atherosclerotic plaque rupture, and even cardiac cell death after occlusion of coronary arteries, are critically regulated by miRNAs [4].

Myocardial ischemic injury is one of the leading causes of myocardial cell death in AMI, and certain cardiac-specific miRNAs critically regulate the ischemic pathway through modification by their targets [5]. Those miRNAs, measurable in various body fluids, including blood, are considered circulating microRNAs. Circulating miRNAs are highly constant in extreme hot and freezing temperatures and exhibit remarkable defensive power against plasma RNAse due to binding with microvesicles and proteins. Therefore, plasma or serum miRNAs fulfill all the criteria as a standard biomarker [6]. Several recent clinical researchers have acknowledged that circulatory miRNA levels are significantly deregulated in patients with atherosclerosis, hypertension, arrhythmia, stable coronary heart disease, unstable coronary heart disease, coronary plaque rupture, acute coronary syndrome, and acute myocardial infarction [2,5,6,7,8,9,10]. Moreover, cardiac-specific circulatory miR-208b and miR-499 were reported as potential clinical biomarkers for acute myocardial infarction patients [2,6,11]. It has been demonstrated that disturbed blood flow and hyperlipidemia surprisingly altered the relative miR-21 expression patterns in cultured human endothelial cells and subsequently enhanced the dysfunction of the endothelial wall and initiation of atherosclerotic plaque formation through several pro-inflammatory targets, including NF-κB (nuclear factor κB)-mediated VCAM-1 (vascular cell adhesion molecule-1) [4,12]. A clinical study reported that miR-21 is prominently involved in coronary atherosclerosis formation in coronary heart disease patients through forkhead box P3 (foxp3) and TGF-β1/Smad-7 signaling pathways [13]. Moreover, altered Smad-7 expressions have a protective role in post-myocardial wound repair through fibroblast activation [14].

Biological aging is considered a major non-modifiable risk factor for myocardial infarction. Dysregulated miR-101-3p, miR-142, miR-146, miR-409-3p, miR-142, and miR-146 expressions have been directly associated with aging [2]. However, the effect of miR-21 in hypoxia-induced H9c2 cellular damage and its relationship with Smad-7 and the aging process are largely unknown. Several researchers documented that miR-21, miR-488, and miR-126 have been linked with cardiovascular diseases, but the diagnostic significance of these miRNAs for the early evaluation of non-ST-segment and ST-segment-elevated acute myocardial infarction patients is not fully explored. Therefore, we assessed the clinical significance of these miRNAs for the early prediction of AMI patients. In addition, we also investigated the molecular impact of cardio-enriched miR-21 in hypoxia-injured H9c2 cells and its association with Smad-7 and aging.

## 2. Section of Material and Methods

### 2.1. Study of Human Research Participants

Our research is a case–control-based, cross-sectional study of 95 non-ST-elevated acute myocardial infarction patients (NSTEMI), 152 ST-elevated acute myocardial infarction patients (STEMI), and 95 healthy subjects, aged between 30–89 years, whom we recruited from February 2022 to July 2023 (n = 342) at the tertiary specialized Xiangya Hospital. NSTEMI and STEMI patients were diagnosed according to ACC/AHA/ESC/WHF Task Force Practice Guidelines [1]. Blood samples were collected within 12 h of acute chest pain. Moreover, AMI patients were diagnosed based on at least two of the following criteria: (I) more than 30 min of acute retrosternal ischemic-type chest pain; (II) ST-segment elevation or depression of more than 1 mm; and (III) elevated plasma cTnI (>0.1 ng/mL). Patients were excluded from our study if they had a known history of previous myocardial infarction, prior intravenous thrombolytic or anticoagulant therapy, arrhythmia, heart failure, percutaneous coronary intervention, coronary artery bypass graft surgery, chronic renal failure, liver failure, or malignant disease. We considered healthy subjects in our study if they had no history of cardiovascular or cerebrovascular diseases and were free from chronic inflammatory diseases. This clinical study was carried out following the medical research principles of the Helsinki Declaration by The World Medical Association and approved by Central South University’s Xiangya first affiliated hospital, and prior to the study, consent was obtained from all the study subjects.

### 2.2. Animal Study Groups

From Xiangya’s Laboratory Animal Center, 12-week-old C57BL/6 male mice were collected. We performed our experiments according to the Guidelines of the Chinese National Institute of Health Care for Laboratory Animals and in accordance with the ethical standards of the Veterinary Center of Xiangya School of Medicine.

A total of 15 mice were divided into three groups of equal sizes: (1) the healthy group, (2) the AMI group, and (3) the sham group (same surgical procedures without coronary artery ligation). AMI was generated as described previously [14,15]. Briefly, mice were kept under general anesthesia by using 100 mg/kg ketamine and 1 mg/kg acepromazine, maintaining proper mechanical ventilation; then, the left anterior descending coronary artery was permanently occluded by the 6/0 silk suture to confirm myocardial ischemia. Blood samples were collected from AMI mice at 12 h of ligated left anterior descending coronary artery (LAD) into a K2-EDTA-coated tube and subsequently obtained pure plasma. Experimental protocols are given in more detail in the flow chart, Figure 1.

### 2.3. H9c2 Cell Culture, Hypoxic Model, Transfection, and Hoechst Staining

From the Shanghai Institute of Health Science, rat ventricular cardiomyocyte-originated H9c2 cells were obtained and grown at 37 °C in 95% humidity and 5% CO_2_ by using Dulbecco’s Modified Eagle Medium and bovine serum, which were replaced every three days by new culture mediums, with approximately 85% of cell growth transferred to the newer flasks. Moreover, the H9c2 hypoxic model was established by exposing the cells to 1% oxygen, 5% CO_2,_ and 95% nitrogen for 12 h at 37 °C using a modular incubator (Model 3131, Forma Scientific, Marietta, OH, USA). Furthermore, H9c2 cells were categorized into (1) the control group (constant maintenance with 37 °C temperature, 95% oxygen, and 5% CO_2_), and (2) the hypoxic group. Moreover, Lipofectamine 2000, miR-21 inhibitor (100 nM), and NC miR-21 (100 nM) reagents were added to H9c2 cells and incubated for 12 h as per the manufacturer’s instruction (Invitrogen, Carlsbad, CA, USA) to develop the transfection. Cellular Hoechst Staining was performed according to the company’s instructions.

### 2.4. Luciferase Reporter Gene Expression

Luciferase assaying technique was conducted to the determination of miR-21 target gene.

The potential target site for miR-21 was predicted using the gene analysis software Target Scan (http://www.targetscan.org/) with the complementary sequence of the 3′ untranslated region of Smad-7. Moreover, the 3′-UTRs of Smad-7 inhibitor miR-21 (5′-UCAACAUCAGUCUGAUAAGCUA-3′), and NC miR-21 (5′-UUCUCCGAACGUGUCACGUT-3′) were transfected into H9c2 cells in 24-well plates using Lipofectamine 2000 for 12 h. The luciferase gene expressions were demonstrated using the Dual-Luciferase Reporter assay reagents, following the manufacturers’ protocols (Beyotime, Shanghai, China), and Renilla was used as an inner control. Fluorescent sensitivity was considered to detect the relative values.

### 2.5. Caspase-3 Activities Assessment

Expressions of caspase-3 levels were demonstrated with caspase-3 activity assay reagents as per the company’s guidelines (Beyotime, Shanghai, China). Firstly, a standard curve was performed using the standard pNA sample from the assay kit. Afterward, 96-well plates of normal and transfected H9c2 cells were lysed through buffer solution, followed by centrifugation at 4 °C and 16,000× *g* for 15 min. Finally, the optical density of each well was assessed at a 450 nm wavelength with a microplate reader (Sunnyvale, CA, USA), and the expressions of caspase-3 activities were calculated compared with controls.

### 2.6. Detection of Intracellular ROS Concentration

As per the company’s instructions, the non-fluorescent probe 2′,7′-dichlorofluorescein diacetate was used (Beyotime, Shanghai, China) to determine the intracellular concentrations of ROS production. Firstly, properly cultured H9c2 cells were transfected by inhibitors and negative controls of miR-21 and inoculated for 12 h in the hypoxic medium. Subsequently, DCFH-DA was added to 96-well plates, inoculated at 37 °C for 15 min, and, afterward, washed with FBS three times. Finally, ROS concentrations were measured with a microplate fluorometer at a 480 nm wavelength (Molecular Devices, Sunnyvale, CA, USA).

### 2.7. Cellular Viability Assay

Cellular survivability was assessed with the Cell Counting Kit-8, following the company’s (Beyotime, China) instruction. After proper growth of H9c2 cells in 96-multi-well plates, subsequently, CCK-8 solutions (10 µL) were added to each well plate of normal, hypoxia-treated, and hypoxia-under-treated miR-21 and incubated for 2 h. Lastly, cellular proliferation was evaluated through an absorbance reader at a 450 nm wavelength (Sunnyvale, CA, USA), with the process repeated four times for each group to ensure validity.

### 2.8. RNA Isolation and Detection of miRNA Expressions

TRIZOL chemicals were added with plasma and also in properly cultured cardiomyocytes as per the company’s instruction (Invitrogen, USA) for obtaining RNA molecules. Bulge-loop reverse transcription microRNA primers were used to reverse-transcribe RNA to cDNA following the company’s instructions (RiboBio, Guangzhou, China). SYBR Green microRNA RT-PCR kits (Takara, Dalian, China) were used to detect the expression levels of miR-21, miR-488, and miR-126 with a quantitative real-time PCR system. The relative expression each miRNA was measured by the comparative Ct (cycle threshold) 2^−ΔCt^ method. MicroRNA-156a was applied to an endogenous control of miR-21, miR-488, and miR-126 expression. Moreover, miRNAs expressions were categorized as high (Ct 15–25), good (Ct 25–32), and undetectable (Ct > 38). More details were explained in our previous study [2,14,15,16]. Target gene primers are shown in Table 1.

### 2.9. Statistical Methods

IBM statistic version SPSS 22 was applied for statistical calculations. Comparisons between groups for continuous variables were performed using one-way ANOVA interpretation, the Mann–Whitney test, or the Student’s *t*-test, with Tukey’s post hoc test for multiple groups. For categorical variables, the Chi-square (*χ*^2^) test or Fischer’s exact test was used as appropriate. All the figures were drawn through GraphPad Prism version 6 and values are expressed as mean ± SEM. Clinical biomarker impact analysis for plasma miRNAs of patients with AMI was carried out through a graphical presentation of receiver operating characteristics (ROCs). A *p*-value < 0.05 was considered statistically significant.

## 3. Results

### 3.1. Baseline Information of the Participants

For the current study, a total of 342 patients were recruited: 95 participants with NSTEMI, including 56 men and 39 women (63.7 ± 11.9 years); 152 patients with STEMI, including 85 men and 67 women (62.4 ± 13.2 years); and 95 healthy volunteers, including 48 men and 47 women (average age: 58.9 ± 12.6 years). The clinical parameters of C-reactive protein, a family history of CAD, ventricular ejection fraction rates, and serum creatinine values were statistically significant between healthy individuals and STEMI and NSTEMI patients but were non-significant between STEMI and NSTEMI subjects. However, all other baseline variables were non-significant among healthy individuals with STEMI and NSTEMI subjects. More details are explained in Table 2.

### 3.2. Expression of Plasma miRNAs (miR-21, miR-488, miR-126) in Acute Myocardial Infarction Patients and Mice AMI

Circulatory miR-21 and miR-488 concentrations were significantly upregulated, while miR-126 expression levels were considerably decreased, both in NSTEMI and STEMI subjects in comparison with healthy volunteers. Moreover, plasma concentrations of miR-21 and miR-488 were slightly higher, and miR-126 concentrations were comparatively lower in STEMI subjects as compared to NSTEMI subjects, but their differences were not statistically significant (Figure 1A–C). We noticed a greatly increased circulatory miR-21 and miR-488 relative expression in our AMI mice model groups compared to control groups. In addition, plasma miR-126 levels were remarkably downregulated in AMI mice groups compared to those of controls. However, the levels of circulatory miR-21, miR-488, and miR-126 among controls and sham groups were not statistically significant (Figure 1D–F).

### 3.3. Diagnostic Impact of Plasma miRNAs for AMI Patients

We performed a ROC curve analysis to explore the diagnostic accuracy of circulatory miR-21, miR-488, and miR-126 for AMI patients. Plasma miR-21 showed a promising potential to discriminate NSTEMI (AUC 0.955) and STEMI (AUC 0.967) patients from control subjects (Figure 2A,B). Moreover, the plasma concentration of miR-488 accurately distinguished NSTEMI (AUC 0.926) and STEMI (AUC 0.934) patients from healthy individuals (Figure 2C,D). Furthermore, circulatory plasma miR-126 clearly separated NSTEMI (AUC 0.825) and STEMI (AUC 0.869) patients from healthy subjects (Figure 2E,F). Therefore, our findings indicate that altered miR-21, miR-488, and miR-126 levels in plasma are novel clinical biomarkers for the early detection of AMI.

### 3.4. Expression and Luciferase Reporter Assay of miR-21, miR-488, and miR-126 in H9c2 Cells

The relative expressions of cardiac-enriched miR-21, miR-488, and miR-126 were upregulated by 2.6-fold and 2.8-fold, respectively, in 12 h hypoxia-exposed H9c2 cells compared to normoxic cells (Figure 3A,B) (*p* < 0.001), while the miR-126 expression level markedly decreased by 5.6-fold in hypoxia-injured H9c2 cells compared to normal cells (Figure 3C). Moreover, a significant decline in miR-21 concentration was noted in hypoxia cells treated with the miR-21 inhibitor compared to untreated cells, and nonsignificant expression was observed within hypoxic and negative controls (Figure 3D). To evaluate the molecular insight of hypoxia-induced cardiomyocyte inflammatory damage linked with alteration of miR-21, we used Target Scan analysis and luciferase gene expression and found that Smad-7 is a target gene of miR-21. Moreover, the luciferase expressions were significantly downregulated in 12 h hypoxia-incubated H9c2 cells compared to normal cells, but the negative control and hypoxic H9c2 cell groups were non-significant. The inhibition of miR-21 markedly regulated luciferase activity in hypoxia-exposed H9c2 cells (Figure 3E).

### 3.5. Effects of Inhibitor miR-21 on Smad-7 and Hypoxia-Exposed H9c2 Cells

The Smad-7 mRNA expression levels were remarkably decreased by 4.1-fold in the hypoxia group as compared to normal H9c2 cells. On the contrary, Smad-7 expressions in hypoxic H9c2 cells treated with the miR-21 inhibitor were significantly elevated by 3.3-fold compared to untreated hypoxic cells, but there was no effect noticed within the negative (NC) control and hypoxic group (Figure 4A). Furthermore, ROS expression levels were obviously upregulated by 4-fold in the 12 h hypoxic group as compared to normoxic H9c2 cells, while ROS levels were prominently down-regulated by two-fold in hypoxia-exposed miR-21-inhibited H9c2 cells compared to only-hypoxia-incubated H9c2 cells. However, there were no changes observed among NCs and hypoxia H9c2 cells (Figure 4B). We revealed that caspase-3 activities were significantly elevated by 9-fold in the hypoxic group compared to normal H9c2 cells, whereas inhibition of miR-21 remarkably reduced hypoxia-induced caspase-3 activities by 7-fold, as compared to hypoxia-injured H9c2 cells, but the expression of caspase-3 levels between the hypoxic group and negative control was non-significant (Figure 4C). In addition, cellular viability greatly decreased by 2-fold in the hypoxic H9c2 group compared to normal controls, whereas cellular survivability increased by 1.9-fold in miR-21-inhibited hypoxic H9c2 cells compared to hypoxic cells (*p* < 0.001) (Figure 4D). The cellular apoptosis rate was calculated through Hoechst staining, and we found markedly elevated cellular apoptosis rates in the hypoxia group compared to the control; in hypoxia-exposed H9c2 cells treated with the miR-21 inhibitor, the cellular apoptosis rate was significantly reduced in the hypoxia group compared to the control (*p* < 0.001) (Figure 4E,F). Therefore, our findings suggest that the inhibition of miR-21 enormously suppressed caspase-3 as well as ROS expression and outstandingly improved cellular viability during hypoxia-induced H9c2 cellular injury by regulating its target Smad-7 mRNA and protein expressions.

### 3.6. The miR-21 Relationship with Aging and Gender

In the older groups (70–89 years), both STEMI and NSTEMI subjects significantly upregulated plasma miR-21 levels compared to younger (30–49 years) patients. In the elder patients’ group (50–69 years), miR-21 expression was reasonably higher than in the younger groups (30–49 years) but was statistically insignificant (Figure 5A,B). In healthy volunteers, miR-21 expressions slowly rose with increasing age. However, the volume of plasma miR-21 between male and female subjects in either healthy or disease groups was statistically insignificant. (Figure 5C). This finding, attributed to miR-21, has a strong link to aged AMI patients, suggesting that low miR-21 levels may prevent the aging process, as well as treat AMI subjects.

## 4. Discussion

Sudden unexpected cardiac death predominantly occurred due to acute myocardial infarction (AMI). Accurate and timely diagnosis of AMI can guarantee urgent initiation of reperfusion treatment that directly reduces the mortality rate. Several clinical and animal studies have suggested that selected cardiac-enriched circulating miRNAs such as miR-1, miR-133, and miR-223 have a significant diagnostic role in the evaluation of AMI patients [2,6,10]. In our study, we observed markedly upregulated circulatory plasma concentrations of miR-488 and miR-21, while finding potentially downregulated miR-126 levels in both the NSTEMI and STEMI group compared to healthy participants. Recently some researchers acknowledged that miR-21 was critically involved in coronary stenosis and highly upregulated in AMI subjects, as well as in hypoxia-injured cardiomyocyte (H9c2) cells; they recommended circulatory miR-21 as having a significant diagnostic impact on acute MI patients, which is consistent with the results of our study [17,18]. Furthermore, circulatory concentrations of plasma miR-21 were noticeably more upregulated in our AMI mice models than in controls. Significantly elevated miR-21 expression patterns were also demonstrated by Yang et al. in mouse models of AMI, and miR-21 prevents excessive inflammation and cardiac dysfunction after myocardial infarction through targeting kelch repeat and BTB (POZ) domain-containing 7 (KBTBD7), considered a new therapeutic molecule for MI; these results are also partially associated with our results [18,19]. It has been recognized that miR-21 is greatly involved in vascular inflammation, foam cell formation, atherosclerosis plaque formation, and also controlling cardiac fibrosis through Smad-7 [20,21]. In our H9c2 cellular study, to find out the underlying molecular mechanism of miR-21 in protecting against hypoxia-induced cardiac cell damage, we performed Target Scan and luciferase reporter gene analysis and confirmed that Smad-7 acts as a potential target gene for miRNA-21. We found that ROS, caspase-3, irreversible cellular damages, and concentrations of miR-21 were markedly elevated, whereas the relative expression of Smad-7 was evidently reduced in 12 h hypoxia-incubated H9c2 cells as compared to normal culture cells. In contrast, Smad-7 expressions were prominently elevated and significantly diminished cellular ROS concentrations and caspase-3 levels, while considerably amplifying the viability of hypoxic H9c2 cells treated with the miR-21 inhibitor. We recommend that the knockdown of miR-21 has significant potential to prevent cardiac cell death through regulating its target, Smad-7. Moreover, recent hypoxia-injured ventricular H9c2 cell studies also demonstrated that markedly elevated expressions of miR-21 increased caspase-3 activity, ROS generation, and cellular death, while hypoxic H9c2 cells treated with the miR-21 inhibition significantly reduced intracellular ROS and caspase-3 levels and remarkably augmented cardiac cellular viability by regulating PDCD4, Akap8, and Bard1 expressions through modification of Wnt/β-catenin, PTEN/PI3K/AKT, and NF-κB pathways [15,22]. These studies, along with our findings, strongly support that miR-21 significantly regulates caspase-3 activity, intracellular ROS generation, and hypoxia-induced H9c2 cellular injury through the modulation of Smad-7 expression.

Our study first recognized the significantly higher expression of plasma miR-488 in acute MI patients and AMI mice models than in controls. Additionally, we detected that miR-488 expressions were remarkably upregulated in hypoxia-exposed H9c2 cells compared to normoxic control groups. Zhen research groups have shown that high expression of circulating miR-488 is a good biomarker for the identification of atherosclerosis patients, and downregulation of miR-488 can inhibit vascular smooth muscle cell (VSMC) proliferation and migration [23]. Verjans et al. reported that macrophages transfected with mimic-miR-488-5p significantly decrease sICAM-1, IL-1ra, IL-16, CXCL1, and TNFα cytokine levels and regulate heart failure conditions in human patients and rat models [24]. However, in our present study, we found that circulatory miR-126 expressions were significantly decreased in both NSTEMI and STEMI patients, AMI mice models, and hypoxia-induced injured H9c2 cells compared to control groups, which is in strong agreement with other recent studies [10,16]. Luo et al. established that expressions of miR-126 levels were prominently decreased in AMI rat models and 24 h hypoxia-exposed H9c2 cell injury. However, overexpression of miR-126 obviously reduced inflammatory markers (TNF-α, IL-6, and IL-1β), apoptosis rates, and cardiomyocyte fibrosis and significantly improved cardiac cell viability and promoted angiogenesis [25].

Moreover, to explore diagnostic accuracy, we performed a receiver operating characteristic (ROC) curves analysis. ROC analysis revealed AUC values for plasma miR-21 in NSTEMI (0.955) and STEMI (0.967), miR-488 in NSTEMI (0.926) and STEMI (0.934), and miR-126 in NSTEMI (0.825) and STEMI (0.869) compared to those of healthy volunteers. These findings strongly suggest that upregulated plasma miR-21 and miR-488 and downregulated miR-126 levels are able to clearly differentiate NSTEMI and STEMI patients from healthy control subjects and could be novel clinical biomarkers for the identification of AMI patients. Furthermore, aging is an independent major risk factor for atherosclerotic acute coronary syndrome, and our study showed that plasma miR-21 was significantly higher in geriatric AMI patients, and other studies also found miR-21 is correlated with cellular aging [2,26,27]. To avoid technical faults related to the PCR method, we analyzed all the samples four times for human, animal, and H9c2 cells with a single investigator, and miRNAs with Ct values less than 15 and more than 38 were not considered in this study, which guaranteed that our findings were more accurate and more reasonable.

## 5. Limitations of the Study

However, our study was only conducted in one provisional multidisciplinary hospital among the Han population with relatively few clinical samples. Therefore, different ethnic populations with larger clinical samples investigations will be required to validate these miRNAs as a useful new biomarker before being applied to clinical practice. Required further study to find out CRP and creatinine’s molecular relationship with our selected miRNAs. Moreover, we did not examine the inflammatory relationship between TGF-β1 and Smad-7, further elucidation of TGF-β1 and Smad-7 signaling pathways is essential to explore their molecular mechanism during hypoxic-induced H9c2 cellular injury. Moreover, we do not know the regular drugs those were taken by AMI patients that were made any effects on miRNAs expressions, and these medications also had any effects on Animal and cell models, need to further research.

## 6. Conclusions

We recommend that augmented miR-21, miR-488, and decreased miR-126 plasma concentrations may play critical roles in the early stage of AMI and may be used as novel clinical predictors for diagnosis of NSTEMI and STEMI patients. In addition, miR-21 has a notable correlation with aging. Inhibition of miR-21 prevents H9c2 cardiac cell death during hypoxic challenges through regulating Smad-7, providing a possible therapeutic strategy for AMI patients.

## Data Availability

Data will be available upon request from the corresponding author.

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
