# Peer review of "MicroRNA-21 Protects Hypoxic-Induced Cardiomyocytes Injury by Targeting Smad-7"

_cells, 2025, doi:10.3390/cells14191483_

Round 1
Reviewer 1 Report
Comments and Suggestions for Authors
In the paper, authors find that upregulated plasma miR-21 and miR-488, and downregulated miR-126 levels might be considered as potential clinical biomarkers for myocardial infarction patients. And by targeting smad-7, the inhibition of miR-21 significantly reduced hypoxic exposed H9c2 cellular injury. This article demonstrates a potentially useful therapeutic target for AMI patients. The work is meaningful and the conclusion is fully supported by all the results. However, there are still some issues that need to be further addressed before its acceptance.
(1) In the Abstract part, the full form of the terms NSTEMI and STEMI should be provided upon their first appearance. This would make it more easily to be understood for readers.
(2) In Table 2, please use integers for age, not decimals.
(3) The flow chart is not presented in a professional and clear fashion. Please check it more carefully.
(4) There are lots of miRNA. And the authors just chose miR-21 and miR-488, and miR-126 for investigation. Why? Is there any previous relevant research results that has been published by the same group?
(5) As presented in Figure 2, the altered levels of miR-21, miR-488 and miR-126 in plasma are new clinical biomarkers for the early detection of AMI patients. But the following research only focus on the effect of miR-21. Why ignored the effects of miR-488 and miR-126? What are the potential targets of miR-488 and miR-126? It needs to be demonstrate whether the regulation of miR-488 and miR-126 also provides a possible therapeutic strategy for AMI patients.
(6) In Figure 1, it is the relative levels of miRNAs that are compared among different experimental groups. What is the standard value that is used for the normalization.
Author Response
Response to the Reviewers comments -1
In the paper, authors find that upregulated plasma miR-21 and miR-488, and downregulated miR-126 levels might be considered as potential clinical biomarkers for myocardial infarction patients. And by targeting smad-7, the inhibition of miR-21 significantly reduced hypoxic exposed H9c2 cellular injury. This article demonstrates a potentially useful therapeutic target for AMI patients. The work is meaningful and the conclusion is fully supported by all the results. However, there are still some issues that need to be further addressed before its acceptance.
Response: Thank you so much
(1) In the Abstract part, the full form of the terms NSTEMI and STEMI should be provided upon their first appearance. This would make it more easily to be understood for readers.
Response: Thank you so much, Non-ST-elevation myocardial infarction (NSTEMI), ST-elevation myocardial infarction (STEMI) provided in abstract section
(2) In Table 2, please use integers for age, not decimals.
Response: Thank you so much, decimals were deleted in aging , in table 2 highlighted with red color
(3) The flow chart is not presented in a professional and clear fashion. Please check it more carefully.
Response: Thank you so much, Flow chart is updated and now is clearer
(4) There are lots of miRNA. And the authors just chose miR-21 and miR-488, and miR-126 for investigation. Why? Is there any previous relevant research results that has been published by the same group?
Response: Our team has already published “Diagnostic Role of Plasma MicroRNA-21 in Stable and Unstable Angina Patients and Association with Aging” (Cardiol Res Pract. 2020 Apr 13:2020:9093151. doi: 10.1155/2020/9093151. eCollection 2020.) and “The mir-21 Inhibition Enhanced HUVEC Cellular Viability during Hypoxia-Reoxygenation Injury by Regulating PDCD4” Mediators Inflamm. 2022 Jun 30:2022:9661940. doi: 10.1155/2022/9661940. eCollection 2022 And miR-488 we are still working on it , and “Overexpression of miR-126 Protects Hypoxic-Reoxygenation-Exposed HUVEC Cellular Injury through Regulating LRP6 Expression”. Oxid Med Cell Longev. 2022 Jan 17:2022:3647744. doi: 10.1155/2022/3647744. eCollection 2022.
(5) As presented in Figure 2, the altered levels of miR-21, miR-488 and miR-126 in plasma are new clinical biomarkers for the early detection of AMI patients. But the following research only focus on the effect of miR-21. Why ignored the effects of miR-488 and miR-126? What are the potential targets of miR-488 and miR-126? It needs to be demonstrate whether the regulation of miR-488 and miR-126 also provides a possible therapeutic strategy for AMI patients.
Response: Thank you so much for your excellent questions, regarding miR-21 and miR-126 we have already published their clinical and therapeutic impact on CAD, currently we have been doing research on miR-488 molecular relationship with AMI. We hope soon we will get our results and will be published. PDCD4 is the potential target for miR-21 and LRP6 is the potential target for miR-126.
(6) In Figure 1, it is the relative levels of miRNAs that are compared among different experimental groups. What is the standard value that is used for the normalization.
Response: Thank you very much, as it is the relative expression ratio therefore it has no specific unit and no specific standard value, we used widely available inner control microRNA-156a for normalization and those miRNA expression Ct (cycle threshold) 2−ΔCt. value Ct (15-25) was used in our results.
Reviewer 2 Report
Comments and Suggestions for Authors
Authors studies miR-21, miR-488, and miR-126 in human, mouse, and cell models of acute myocardial infarction, and found interesting and potentially clinically important information.
Please define abbreviations “NSTEMI” and “STEMI” in the abstract.
Introduction, last paragraph: Please define the abbreviation “ST”.
Methods the first sentence of Section 2.2: Need a comma and a space between “Center” and “12”.
Methods the first sentence of Section 2.2: Please describe if the applications used male mice or female mice.
Authors are all located in Saudi Arabia, but both human and animal studies were performed in China? Please clarify.
Please describe the functions of inhibitor-miR-21 (5'- 138
UCA ACAUCAGUCUGAUAAGCUA-3') and NC-miR-21 (5′-UUCUCCGAACGUG- 139
UCACGUT-3′)
Please expand the information in the Fig. 5 legend.
6. Conclusion section: Please re-word “We recommend that … may play critical roles….”
Could the observed effects in patients may be due to medications taken? What are the effects of these medications on the animal and cell models?
Author Response
Response to the Reviewer’s Report-2
Authors studies miR-21, miR-488, and miR-126 in human, mouse, and cell models of acute myocardial infarction, and found interesting and potentially clinically important information.
Response: Thank you so much for your positive response.
Please define abbreviations “NSTEMI” and “STEMI” in the abstract.
Response: Thank you so much, Non-ST-elevation myocardial infarction (NSTEMI), ST-elevation myocardial infarction (STEMI) provided in abstract section
Introduction, last paragraph: Please define the abbreviation “ST”.
Response: Thank you so much, non-ST-segment and ST-segment-elevated acute myocardial infarction patients
Methods the first sentence of Section 2.2: Need a comma and a space between “Center” and “12”.
Response: Thank you so much, corrected
Methods the first sentence of Section 2.2: Please describe if the applications used male mice or female mice.
Response: C57BL/6 male mice were used in our experiment, Kindly look in 2.2. Animal study groups words are highlighted in red color.
Authors are all located in Saudi Arabia, but both human and animal studies were performed in China? Please clarify.
Response: Thank you so much, I have been completed my PhD from Xiangya hospital of Central South University and after that I was worked there before came to Saudi Arabia as a result, Till I have a strong collaboration with cardiology department specially Prof Yang tiang lun, Professor Xia Ke and Professor Li fei, I used their names in acknowledgements
Please describe the functions of inhibitor-miR-21 (5'- 138
UCA ACAUCAGUCUGAUAAGCUA-3') and NC-miR-21 (5′-UUCUCCGAACGUG- 139
UCACGUT-3′)
Response: Thank you so much, inhibitor-miR-21 (5'- 138 UCA ACAUCAGUCUGAUAAGCUA-3') is used to block or reduce the endogenous miR-21 activities and NC-miR-21 (5′-UUCUCCGAACGUG- 139
UCACGUT-3′) is used for non-coding negative control and it has no biological function.
Please expand the information in the Fig. 5 legend.
Response: Thank you so much, Fig. 5 legend is fully expanded.
- Conclusion section: Please re-word “We recommend that … may play critical roles….”
Response: Thank you so much, corrected
Could the observed effects in patients may be due to medications taken? What are the effects of these medications on the animal and cell models?
Response: Thank you so much, it is great questions, Really, we don’t know the regular drugs those were taken by AMI patients that were made any effects on miRNAs expressions, and these medications also had any effects on Animal and cell models, these are our limitation need to further research.
Round 2
Reviewer 1 Report
Comments and Suggestions for Authors
All the questions raised have been addressed. And the manuscript is fine in the present form.
Reviewer 2 Report
Comments and Suggestions for Authors.